# BAX Redistribution Induces Apoptosis Resistance and Selective Stress Sensitivity in Human HCC

**DOI:** 10.3390/cancers12061437

**Published:** 2020-05-31

**Authors:** Kathrin Funk, Carolin Czauderna, Ramona Klesse, Diana Becker, Jovana Hajduk, Aline Oelgeklaus, Frank Reichenbach, Franziska Fimm-Todt, Joachim Lauterwasser, Peter R. Galle, Jens U. Marquardt, Frank Edlich

**Affiliations:** 1Institute for Biochemistry and Molecular Biology, University of Freiburg, 79104 Freiburg, Germany; kathrin.funk@biochemie.uni-freiburg.de (K.F.); ramona.klesse@biochemie.uni-freiburg.de (R.K.); aline.oelgeklaus@sgbm.uni-freiburg.de (A.O.); frank.reichenbach@sgbm.uni-freiburg.de (F.R.); franziska.todt@biochemie.uni-freiburg.de (F.F.-T.); joachim.lauterwasser@biochemie.uni-freiburg.de (J.L.); 2Faculty of Biology, University of Freiburg, 79104 Freiburg, Germany; 3Department of Medicine, Lichtenberg Research Group, University Mainz, 55116 Mainz, Germany; cczauder@uni-mainz.de (C.C.); Beckerdi@uni-mainz.de (D.B.); hajdukjovana@gmail.com (J.H.); peter.galle@unimedizin-mainz.de (P.R.G.); 4Spemann Graduate School of Biology and Medicine (SGBM), University of Freiburg, 79104 Freiburg, Germany; 5Department of Medicine I, University Hospital Schleswig-Holstein, Ratzeburger Allee 160, 23562 Lübeck, Germany; 6CIBSS Centre for Integrative Biological Signalling Studies, University of Freiburg, 79104 Freiburg, Germany

**Keywords:** BCL-2 family, BH3, mitochondrial apoptosis, DNA damage, BH3 profiling

## Abstract

Cancer therapies induce differential cell responses, ranging from efficient cell death to complete stress resistance. The BCL-2 proteins BAX and BAK govern the cellular decision between survival and mitochondrial apoptosis. Therefore, the status of BAX/BAK regulation can predict the cellular apoptosis predisposition. Relative BAX/BAK localization was analyzed in tumor and corresponding non-tumor samples from 34 hepatocellular carcinoma (HCC) patients. Key transcriptome changes and gene expression profiles related to the status of BAX regulation were applied to two independent cohorts including over 500 HCC patients. The prediction of apoptotic response was tested using cell lines and polyclonal tumor isolates. Cellular protection from BAX was confirmed by challenging cells with mitochondrial BAX. We discovered a subgroup of HCC with selective protection from BAX-dependent apoptosis. BAX-protected tumors showed enrichment of signaling pathways associated with oxidative stress response and DNA repair as well as increased genetic heterogeneity. Gene expression profiles characteristic to BAX-specific protection are enriched in poorly differentiated HCCs and show significant association to the overall survival of HCC patients. Consistently, addiction to DNA repair of BAX-protected cancer cells caused selective sensitivity to PARP inhibition. Molecular characteristics of BAX-protected HCC were enriched in cells challenged with mitochondrial BAX. Our results demonstrate that predisposition to BAX activation impairs tumor biology in HCC. Selective BAX inhibition or lack thereof delineates distinct subgroups of HCC patients with molecular features and differential response pattern to apoptotic stimuli and inhibition of DNA repair mechanisms.

## 1. Introduction

Hepatocellular carcinoma (HCC) ranks among the most common and rapidly evolving cancers in the Western world [1]. The majority of HCCs emerge as the consequence of chronic inflammatory liver damage that promotes dysregulation of multiple cellular signaling pathways [2]. Herein, alterations of the hepatic microenvironment induce changes of the balance between proliferation and cell death, which overall predisposes malignant transformation. Therefore, failure of immunological clearance of damaged pre-neoplastic hepatocytes as well as the emergence of resistance to death signaling can be considered hallmarks of hepatocarcinogenesis [3]. The molecular profile of resulting HCCs frequently reflects the underlying type, and the magnitude of cell damage with consecutive induction of cytotoxic stress response. The resulting heterogeneity with diverse genetic alterations and various different phenotypes also induces adaptive changes that confer properties of chemoresistance in cancer cells and, therefore, impairs the development of effective therapeutic strategies [2].

Cytotoxic cell stress can induce apoptosis dependent on the proapoptotic BCL-2 proteins BAX and BAK [4]. The activation of BAX and BAK induces permeabilization of the outer mitochondrial membrane (OMM) and activation of the caspase cascade [5]. Human cells protect themselves from BAX/BAK activity by constant shuttling of BAX and BAK from the mitochondria into the cytosol [6,7,8]. This BAX/BAK retrotranslocation depends on prosurvival BCL-2 proteins [8,9]. The retrotranslocation rate determines the amount of mitochondrial BAX/BAK and cellular apoptosis response [8,9]. BAX shuttling is typically faster than BAK retrotranslocation in cultured cells. Therefore, BAX resides predominantly in the cytosol and BAK on the mitochondria. In addition, prosurvival BCL-2 proteins can sequester “activator” BH3-only proteins, thereby preventing BH3-only protein mediated activation of BAX and BAK [10,11,12,13,14,15]. The BCL-2 protein interplay can be characterized by the titration of BH3 mimetic peptides in lysed cells, termed “BH3 profiling”, to identify the most suited BH3 mimetic to induce apoptosis in a tumor [16,17]. Apoptotic response to cell stress and therapeutic outcome can be predicted by the analysis of relative BAX/BAK localization [18]. Furthermore, changes to BCL-2 protein regulatory network and compensatory emergence of antiapoptotic effectors are among the most prominent changes during liver cancer development [19]. However, the molecular underpinnings as well as the subsequent molecular alterations induced by disruption of BCL-2 family members including potential clinical and therapeutic implications remain elusive in HCCs.

We here demonstrate that changes to the localization equilibrium of BAX are frequent during hepatocarcinogenesis. Resulting resistance to cytotoxic stress divides tumors into BAX-protected and non-protected HCCs with differential sensitivity towards apoptotic stimuli. While non-protected HCC display disruption of mitochondrial function and high levels of mitochondrial BAX protected HCC show predominant cytosolic BAX localization and experience elevated oxidative stress. Furthermore, higher proliferative capacity renders non-protected HCC sensitive to classical chemotherapy. In contrast, continuous cellular stress and increased DNA damage with subsequent activation of DNA repair mechanisms led to increased susceptibility to PARP-inhibitors in BAX-protected HCCs. 

## 2. Results

### 2.1. HCCs Show Selection towards Low Mitochondrial BAX Levels

We first analyzed whether BAX/BAK regulation determines the natural course of disease in HCC. Therefore, relative BAX/BAK localization was measured in a cohort of HCC tumor isolates and corresponding non-tumor tissue (Figure 1A). Relative mitochondrial (BAX: cyan; BAK: yellow) and cytosolic protein (BAX: Red; BAK: Orange) were combined to relative protein localizations (relative BAX localization (blue); relative BAK localization (green)) to cross-compare samples as described previously [18]. High values point to increased mitochondrial BAX/BAK localizations and, thus, increased apoptosis predisposition. The results show a strong correlation between BAX and BAK localization in tumor and non-tumor samples (Figure 1B). The BAX localization shows a slight inverse correlation to total protein levels in the tumor consistent with human AML [18]. However, the opposite tendency is apparent in corresponding non-tumor cells (Figure 1C and Appendix A). BAX is shifted towards the cytosol of tumors compared to BAK (Figure 1D). This shift is expected based on the differential localization of both BCL-2 proteins in cell culture [5]. Surprisingly, no significant difference between BAX and BAK can be detected in non-tumor tissue (Figure 1E). These results suggest that BAX and BAK retrotranslocate at similar rates in non-tumor cells. Therefore, the establishment of the differential BAX/BAK localization seems to occur during malignant transformation and result from specific acceleration of BAX retrotranslocation.

Mitochondrial BAX in non-tumor tissues associates with strong cytosolic shifts to corresponding tumors (Figure 1F). Uniform low mitochondrial BAX level in HCC cells point to a specific selection for cytosolic BAX (Figure 1G and Appendix A). By contrast, BAK shows no tendency to shift to the cytosol in tumors but indicates stress-induced shifts to the mitochondria (Appendix A). The importance of limiting the mitochondrial BAX pool is particularly apparent for a subgroup of tumors with high mitochondrial BAX levels in corresponding non-tumor cells (Figure 1H,I and Appendix A). This subgroup is selected for minimizing the mitochondrial BAX pool manifested in a predominant cytosolic BAX localization in tumor cells to protect itself specifically from BAX activation. The selection yields in a BAX localization shift typically of about 2 log scales to the cytosol, regardless of the apoptosis predisposition in non-tumor cells and BAK regulation. Therefore, these tumors will be referred to as BAX-protected HCC.

### 2.2. Distinct Molecular Alterations Drive Cancer Progression in BAX-Protected HCCs

The regulation of the BAX localization could not only influence overall apoptosis predisposition, but also differential sensitivity to apoptotic stimuli. Key transcriptome changes and specific signaling pathways affected in BAX-protected and non-protected tumors regarding their BAX regulation were identified. A total of 66 genes were differentially expressed between BAX-protected and non-protected tumors in non-tumor tissue and 47 genes in tumor tissue. The results suggest a common molecular underpinning of selective BAX inhibition (Figure 2A). Further, Ingenuity Pathway Analyses demonstrated that differences in key signaling pathways and gene sets affected in non-tumorous tissue between BAX-protected tumors and non-protected tumors included mitochondrial dysfunction and apoptosis, whereas changes in tumor tissues were associated with proliferation, oxidative stress response and DNA damage (Figure 2B,C). Therefore, cellular BAX localization was characterized by key changes in apoptosis resistance within the hepatic microenvironment that potentially predisposed malignant transformation. Resulting tumors showed distinct patterns of molecular changes in hallmark oncogenic signaling validating the functional differences in BAX regulation and the tumor classification. To investigate whether the differential BAX localization has a potential impact on biological traits of tumors, we integrated our results with two independent cohorts of authentic HCC patients and assessed clinical outcomes by sub-clustering the tumors based on BAX-protected gene expression signatures from non-tumor and tumor tissues. Notably, survival data was only available for the minority of patients in the corresponding cohort used to establish the signatures due to short or lost follow-up, thus preventing detailed clinical analyses in this cohort. However, a significant association to overall survival of patients could be revealed for both the non-tumor (*p*-value < 0.0001) as well as tumor (*p*-value = 0.0014) signature in both independent patient cohorts (Figure 2D) [20,21]. Moreover, gene set enrichment analyses confirmed that protection associated gene expression signatures during malignant transformation were highly enriched in patients with poor differentiation and adverse clinical outcome represented by the well-established prognostic subclass of tumors, i.e., the subtype A defined by Lee et al. (Figure 2D,E) [21]. Thus, protection against BAX-dependent apoptosis is characteristic for a subclass of patients with adverse tumor features and poor clinical outcome.

### 2.3. BAX-Protected HCC Upregulate DNA Damage Repair

Next, we individually compared the transcriptome profile between non-tumorous and tumor tissue to dissect differences during malignant transformation and identify potential molecular drivers in BAX-protected cancers. A total of 726 genes in the subgroup of BAX-protected tumors and 2029 genes in the subgroup of non-protected tumors showed differential regulation in tumor vs. non-tumorous tissue, effectively separating tumor from non-tumorous tissue in both groups (Figure 3A). Major cellular processes in non-protected tumors involved activation of metabolic processes and protein synthesis (Figure 3B, Table 1) and centered around apoptosis, proliferation, and stress response pathways (Figure 3C). Further, this group showed enrichment in gene sets involved in mTOR and MYC signaling (Appendix A). In contrast, BAX-protected tumors showed downregulation of apoptotic signaling, but activation of DNA replication, recombination, and repair pathways (Figure 3C, Table 1). In addition, enrichment of gene sets associated with DNA damage as well as BRCA signaling was observed (Appendix A). Accordingly, a trend for increased genetic alterations has been observed in BAX-protected tumors (Figure 3D,E).

### 2.4. Mitochondrial BAX Induces Effects Mimicking Selection of BAX-Protected HCC

Next, we tested whether cells challenged with mitochondrial BAX would display a BAX-protected gene expression pattern using HCT116 BAX/BAK DKO cells devoid of intrinsic BAX and BAK [22] challenged with either BAX S184E predominantly localized in the cytosol or BAX S184V with largely mitochondrial localization (Appendix A). Transient expression of both BAX variants confirms that cells with predominant mitochondrial BAX show increased apoptosis predisposition and are therefore exposed to increased selection pressure (Figure 4A and Appendix A). The BAX-protected gene signature suggests a strong dependence on DNA damage repair. We, therefore, analyzed the response of cells to induced DNA damage by DNA-intercalating Daunorubicin and inhibition of DNA damage repair by the FDA-approved inhibitor of the poly ADP ribose polymerase (PARP) Olaparib alone and together. In parallel to BAX-protected HCCs, establishment of stable BAX S184V expression results in reduced apoptosis predisposition (Figure 4B,C and Appendix A). Consistently, gene expression profiles of BAX S184V-challenged cells showed activation of mitochondrial networks and DNA repair as well as apoptosis and proliferation (Appendix A). Furthermore, in accordance with HCC patients, HCT116 cells with cytosolic BAX expression, representing the non-protection subgroup, showed enrichment of gene sets involved in apoptosis as well as cell cycle. The analysis of transiently vs. stably expressed wild type BAX, however, does not reveal a clear difference in localization (Figure 4D,E, Appendix A). The precise analysis of the mitochondrial BAX pool by carbonate extraction separating BAX in loose association with the mitochondria in the carbonate supernatant from OMM-integral BAX in the carbonate pellet shows a prominent shift (Figure 4F,G, Appendix A). Stably expressed wild type BAX is shifted significantly to the mitochondria-associated fraction, suggesting selection towards minimizing the membrane-integral BAX pool. These results validate that challenging cells with mitochondrial BAX resembles the selection of BAX-protected HCC overall confirming the importance of BAX localization for subclassification and therapeutic implications of HCC patients.

### 2.5. BAX-Specific Inhibition Increases Sensitivity to Inhibition of DNA Damage Repair

To validate a functional role of BAX/BAK regulation in primary cell lines, cultured tumor isolates from three different patients were analyzed [23]. Although reflecting largely the tumors, these polyclonal isolates underwent significant changes towards similar BAX/BAK regulation after initial cell culturing (Figure 5A and Appendix A). However, the link between intracellular BAX shuttling and sensitivity to apoptosis induction remains apparent (Figure 5B). In parallel, established hepatoma cell lines reflect only a small fraction of BAX/BAK regulation present in human cells (Figure 5C). Despite genetic differences, HUH7 cells and Hep3B cells display similar relative BAK localization. Therefore, differences in their mitochondrial apoptosis regulation are dependent on BAX. HUH7 cells with largely cytosolic BAX show low and delayed induction of caspase 3/7 activity by kinase inhibitors and topoisomerase inhibitors, whereas Hep3B cells with increased mitochondrial BAX pools show strong apoptotic response (Figure 5D, Appendix A). Targeting the BCL-2 family with the BH3 mimetics ABT-737 and UMI-77 did not induce differential response alone, suggesting apoptosis predisposition is independent of BCL-2 protein levels (Appendix A).

Next, differentially regulated pathways in BAX-protected vs. non-protected tumors were targeted. To this end, HUH7, Hep3B, and HCC68 cells were tested, because they reflect the full range of BAX regulation in cultured hepatoma cells but show similar BAK localization, connecting differential apoptotic responses to BAX regulation (Figure 5C). Upregulation of mTOR signaling in non-protected tumors suggests increased susceptibility to the mTOR inhibitor rapamycin. Among the tested cell lines, HCC68 with predominant mitochondrial BAX show the highest sensitivity towards rapamycin and HUH7 cells with largely cytosolic BAX show the lowest apoptosis response with significantly less caspase activation (Figure 5E). Based on the increased activation of DNA repair in BAX-protected tumors, the PARP inhibitor Olaparib was used (Figure 5F). HUH7 cells are sensitive to 10 µM and higher concentrations of Olaparib with significant increase in apoptosis and reduced colony formation. Notably, HUH7 sensitivity to Olaparib occurs despite insensitivity to a large range of cell stresses (Figure 5D, Appendix A). Therefore, HUH7 cells reflect BAX-protected tumor cells, whereas Hep3B cells and HCC68 cells show characteristics of non-protected tumor cells. It should be noted that these differences originate not necessarily from the present relative BAX/BAK localization, but from processes during their tumorigenesis. Combination of Olaparib with DNA-intercalating Daunorubicin induces synergistic caspase activation further emphasizing the sensitivity of HUH7 cells to DNA repair inhibition (Figure 5G). Olaparib-dependent enhancement of Daunorubicin-induced apoptosis is particularly remarkable, given that apoptosis response to Olaparib alone occurs delayed. Interestingly, Olaparib administration induces a shift of BAX towards the cytosol (Figure 5H,I and Appendix A). However, BAX localization in Olaparib-sensitive HUH7 cells is restored after 24 h, while Hep3B and HCC68 cells maintain their BAX localization, suggesting lower stress response capacity in BAX-protected cells. Together, these results show that BAX-protected cells show higher susceptibility to DNA repair inhibition by Olaparib.

## 3. Discussion

Resistance to cell death is a recurrent hallmark of many cancers including primary liver cancer [24]. We here demonstrate regulation of BAX, and to a lesser extent of the functionally redundant BAK, within the diseased hepatic microenvironment during malignant transformation. In a subgroup of patients, BAX shifts to the cytosol accompanied by fundamental molecular changes in the tumor cells. Despite common cellular control of BAX and BAK particular low mitochondrial BAX levels are apparent in tumor cells (Figure 1). Our analysis identifies increased apoptosis predisposition through mitochondrial BAX prior hepatocarcinogenesis as a potential driver. Therefore, an increased apoptotic predisposition prior tumorigenesis could result in selection towards a BAX-protected tumor. Differences in the equilibrium between cytosolic and mitochondrial BAX delineates distinct subgroups of HCC patients and induces subsequently/as a downstream effect substantial changes in the transcriptome profile of respective patients. Challenging cells with mitochondrial BAX S184V supports these results (Figure 4). As BAX localization is dependent on BAX retrotranslocation, an increased presence of prosurvival BCL-2 proteins on the mitochondria could contribute to this process. However, it is not obvious how altered prosurvival BCL-2 protein levels on the mitochondria would inhibit BAX selectively and not both proapoptotic BCL-2 proteins. On the other hand, BAX mutants, for instance, of the C-terminal TMD, are not common in tumors. Therefore, the molecular underpinning of selective BAX inhibition in tumors will be an important subject of future research.

Strikingly, a significant shift between the localizations of BAX and BAK is not apparent in non-tumor tissue, contrasting the differential BAX/BAK localizations in cultured cells [25,26,27,28,29]. BAX and BAK shuttle between cytosol and mitochondria with different retrotranslocation rates, manifesting differential steady state localization [6,7,8]. However, BAX/BAK regulation in human patients has a much wider variety of scenarios. Tumorigenesis of a subgroup of tumors could establish differential regulation of BAX and BAK, perhaps through stress and BH3-only protein-dependent signaling [16,30]. In addition to a significant resistance to apoptotic stimuli, BAX-protected tumors show increased chromosomal instability with subsequent oncogenic dependence on DNA repair (Figure 5). This is mirrored when cells are challenged with mitochondrial BAX. 

BAX/BAK-dependent apoptosis induction during chronic liver damage in response to oncogene activation, DNA damage, and senescence is well-established and a key mechanisms of cancer prevention [31]. In contrast, evasion of cell death and proapoptotic stimuli in non-transformed hepatocytes is frequently observed during liver cancer development and accompanied by downregulation of BAX [32]. We here demonstrate that transformation of diseased hepatocytes favors enhanced BAX retrotranslocation, identifying a previously unrecognized mechanism during hepatocarcinogenesis that is independent of direct transcriptional regulation. Differences in the equilibrium between cytosolic and mitochondrial BAX delineates distinct subgroups of HCC patients and induces substantial changes in the transcriptome profile of respective patients. BAX-protected tumors show activated oxidative stress and DNA damage suggesting that alterations within the hepatic microenvironment potentially predispose malignant transformation of HCC. These observations are in accordance with the causative role of inflammatory cell death in disease development and progression observed in the majority of HCC [33]. Consistently, GSEA analyses demonstrated that the BAX-protected profile was highly enriched in a prognostic subtype of 139 patients with aggressive tumor biology and poor differentiation [21]. Furthermore, integrative molecular analyses confirmed the prognostic impact of transcriptomic changes that conferred protection against apoptotic stimuli. Therefore, different therapeutic strategies might be required according to the apoptosis regulation of patients to induce treatment effect. While metabolic changes and classical oncogenic pro-proliferative signaling in the non-protection group suggests response to classical chemotherapy and MTOR inhibitors, the dependence on DNA repair and “BRCAness” in BAX-protected tumors might confer high sensitivity to PARP inhibitors. Consistently, HUH7 cells resembling BAX-protected tumors showed selective sensitivity to PARP inhibition by Olaparib. 

## 4. Materials and Methods

### 4.1. Quantification of Western Blot Data

Blots for quantification were detected with LAS400 CCD camera or Vilber Lourmat Solo S, quantification was performed using ImageJ. Proteins were normalized using a titration of the same HeLa whole cell lysate standard curve within each individual blot; BAX (E63, Abcam, Cambridge, MA, USA) and BAK (EMD Millipore, Billerica, MA, USA) were then quantified using SigmaPlot™. The HeLa cell lysate serves as standard protein mix to provide consistent amounts of all analyzed proteins for of Western blot standardization. The amounts of mitochondrial and cytosolic protein were then determined as relative to COX IV (Invitrogen) and β-ACTIN (Millipore), respectively. Finally, ratios were built from the mitochondrial and cytosolic values to obtain the relative protein localization, to allow comparison of samples analyzed not on the same blot, relative mitochondrial and relative cytosolic protein was determined using the fractionation loading controls COXIV and Actin. 

### 4.2. Patient Data 

Tissue from 34 patients with confirmed HCC undergoing resection at the Department of Surgery, University of Mainz, Germany were collected following patient informed consent and local ethics committee approval. This research has been approved by the ethic committee of the Landesärztekammer Rheinland-Pfalz on 05/15/13 (ethic code: 837.199.10 (7208). Detailed description of the cohort can be found [33]. Total RNA was extracted using the Qiagen RNEasy mini Kit (Qiagen GMBH, Hilden, Germany) following the manufacturer’s instructions. RNA quantity and purity were estimated using a Nanodrop ND-1000 Spectrophotometer (NanoDrop Technologies, Wilmington, DE, USA), and integrity was assessed by Agilent 2100 Bioanalyzer (Agilent, Palo Alto, CA, USA). DNA was extracted using Qiagen Qiamp DNA Kit (Qiagen GMBH, Hilden, Germany) following the manufacturer’s instructions.

### 4.3. Subcellular Fractionation

Hepatocellular tissue patient samples (tumor and surrounding tissue) were extracted during surgery and samples were immediately shock frozen and stored in liquid nitrogen. Samples were thawed on ice and washed with ice-cold 1 × PBS. Tissues and cells were resuspended in SEM buffer (10 mM HEPES, 250 mM sucrose, pH 7.2) supplemented with protease inhibitors for 20 min on ice and homogenized using the MINILYS (PEQLab, Erlangen, Germany) glass mill system. Subsequently, samples were centrifuged at 1500× *g* for 5 min at 4 °C. The supernatant was transferred with a 30 G cannula to a 1.5 mL tube and centrifuged for 20 min at 13,000× *g* at 4 °C. While sedimented mitochondria were washed with SEM buffer, the supernatant was ultracentrifuged at 150,000× *g* for 1 h at 4 °C to obtain the cytosolic fraction. Finally, the samples were prepared in SDS sample buffer and separated by SDS-PAGE (NuPage Novex 4–12% Bis-Tris Midi) and subjected to Western blot analysis.

### 4.4. Carbonate Extraction

Mitochondrial pellets were resuspended in 100 mM Na_2_CO_3_ at pH = 11.5 and incubated on ice for 20 min. Subsequently, membranes were pelleted at 15,000× *g* for 30 min at 4 °C. The supernatant, OMM-associated proteins, was subjected to protein precipitation by acetone. The pellet resuspended once more in 100 mM Na_2_CO_3_ at pH = 11.5, incubated on ice for 20 min and centrifuged at 15,000× *g* for 30 min at 4 °C to obtain OMM-integral proteins. Both fractions were assayed by Western blot.

### 4.5. Gene Expression Analysis

Expression values were extracted from Gene Expression Omnibus database (http://www.ncbi.nlm.nih.gov/geo, accession number: GSE84598). Gene expression values were normalized by quantile normalization method across all samples following subtraction of background noises in each spot by GenomeStudio (Illumina^®^). Signal intensity with a detection *p* > 0.05 was treated as a missing value, and only genes with sufficient representation across the samples were included in further data analysis. Differentially expressed genes were determined using Student’s *t*-test included in R version 3.3.3.

Hierarchical cluster analyses were based on Pearson Correlation and complete-linkage clustering, including a filter of 80% presence for each gene. Results were visualized with Complexheatmap (version 1.12.0) [34]. Ingenuity Pathway Analysis (Ingenuity Systems Inc.) tools were used for functional classification and network analyses. The significance of each network, function and pathway was determined by the scoring system provided by Ingenuity Pathway Analysis tool. Gene Set Enrichment analysis (GSEA) was performed using GSEA software provided by Broad Institutes (http://www.broad.mit.edu/gsea/). Gene sets with a NOM *p*-value < 0.05 and FDR < 0.25 were considered significantly enriched in a priori defined set of genes.

### 4.6. Genomic Analyses

Genomic analyses were performed using GenomeStudio (version 2011.1). All SNVs with call Frequencies > 0.95 and Gen Train Score > 0.7 were included for further analyses. CNV were detected by cnvPatition (version 3.2.0) Plugin GenomeStudio. LOH were detected using Bioconductor package VegaMC (version 3.22.0) [35].

### 4.7. Caspase 3/7 Assay and PARP Cleavage

Cells were treated according to the outlined protocol. Whole-cell lysates were prepared as described and incubated with Caspase 3/7 substrate (BD Pharmingen) for 60 min at 37 °C in a 96-well plate (OptiPlate, Perkin Elmer) and protein concentration was determined by a Bradford Assay (Roth). Substrate cleavage was measured for 50 cycles with 10 s delay (excitation at 380 nm, emission at 430–460 nm, Victor X4, Perkin Elmer). Kinetics were measured and calculated to the amount of protein per sample. Cells were treated according to the outlined protocol. For PARP cleavage analysis the samples were precipitated by acetone, boiled in 3× SDS-sample buffer for 10 min at 95 °C, and subjected to SDS-PAGE and Western blot analysis, using anti-Actin C4 (Sigma) and PARP polyclonal (Cell Signaling).

### 4.8. Clonogenic Survival Assay

Cells were seeded to 70% confluency and treated with indicated concentrations of Olaparib (72 h). Medium was exchanged and cells were allowed to recover for 24 h, then cells were splitted and transferred to a new 6-well plate. Cells were cultured for 10–12 days followed by fixation with 4% PFA in 1× PBS and staining with 1% Methylene Blue in 50% methanol. Colonies were identified and counted using ImageJ software.

### 4.9. Cell Culture and BAX Expression

HeLa cell lines were cultured in DMEM 1 g/L glucose medium supplemented with 10 mM Hepes and 10% heat-inactivated fetal bovine serum in 5% CO_2_ at 37 °C. Hep3B, HepG2, and HuH7 cell lines were cultured in DMEM (Gibco, Thermo Fisher Scientific Inc.) medium supplemented with 10 mM Hepes (BioWest), 10% heat-inactivated fetal bovine serum (Biochrom, Schaffhausen, CH), and 1 g/L glucose in 5% CO_2_ at 37 °C, and the patient-derived cell lines (HCC9, HCC31, and HCC68) were supplemented with 5% heat-inactivated fetal bovine serum. Mammalian cell were seeded on 15 cm dishes and harvested at 80% confluency or indicated time of treatment with a cell scraper in ice cold 1× PBS, transferred to a 50 mL falcon tube and cells were pelleted by centrifugation (1200× *g* for 5 min at 4 °C). Cells were regularly (in 4 week intervals) tested for potential mycoplasma infection using the Venor GeM kit (Biochrome).

BAX constructs were cloned into the pEGFP.C3 vectors that were kindly provided from RJ. Youle Lab, NIH, NINDS, USA, the BAX C-terminal serine 184 residue was substituted with valine residue to direct BAX to the mitochondria or with glutamic acid to direct BAX towards the cytosol. HCT 116 BAK^−/−^ BAX^−/−^ cells were cultured in McCoy’s 5A medium supplemented with 10% FCS and 1 mM HEPS at 37 °C in 5% CO_2_. Cells were transfected with Turbofect (Thermo Fisher Scientific Inc.) with the mutation bearing constructs of BAX, according to the manufacturer’s instructions. The day before the experiment HCT116 DKO cells were seeded into the 6-well plates or 15 cm dishes, which had the 70–80% of confluency at the time of transfection. G418 (1000 µg/mL) was added and refreshed every second day, to induce stable transfection. Cells were treated for 4 h or 24 h to induce apoptosis and analyzed by Caspase-3/7 activity assay. 

### 4.10. Preparation of Whole-Cell Lysates

HeLa cell lysate was used to establish a standardized mix of all analyzed proteins, ensuring similar standardization of all patient samples. Cells were harvested, washed with ice-cold 1× PBS and subsequently resuspended in cell lysis buffer (20 mM Tris, 100 mM NaCl, 1 mM EDTA, 0.5% Triton X-100, pH 7.5) supplemented with protease inhibitors. Upon incubation on ice for 15 min, the samples were centrifuged at 15,000× *g* for 10 min at 4 °C. The supernatants were subjected to acetone precipitation, followed by resuspension in SDS sample buffer and storage at −80 °C.

### 4.11. Statistics, Databases and Patient Integration

Statistical analysis was performed using Student’s *t*-test, Friedman test for multiple group comparisons, followed by Dunn’s post hoc test or one-way ANOVA using Holm–Sidak method as indicated. *p*-values ≤ 0.05 were considered statistically significant. Results are presented as means ± SD or means ± SEM as indicated. Survival analyses were performed using log rank (Mantel-Cox) tests.

For integration of patients, publically available expression data sets were used [36]. Hierarchical cluster analyses were performed using Euclidean distance by Bioconductor package multiClust (version 1.4.0). Missing values were computed by k-Nearest Neighbour Imputation with CRAN package VIM (version 4.7.0) [37]. Survival analyses were performed by CRAN package survival und survminer (version 0.4.3) using log rank (Mantel-Cox) tests.

## 5. Conclusions

Together, we discover a previously unrecognized link between cellular BAX localization and apoptosis resistance that characterizes a differential mechanism of malignant transformation in affected patients and links adverse tumor biology to high genetic instability and poor outcome. 

## Figures and Tables

**Figure 1 cancers-12-01437-f001:**
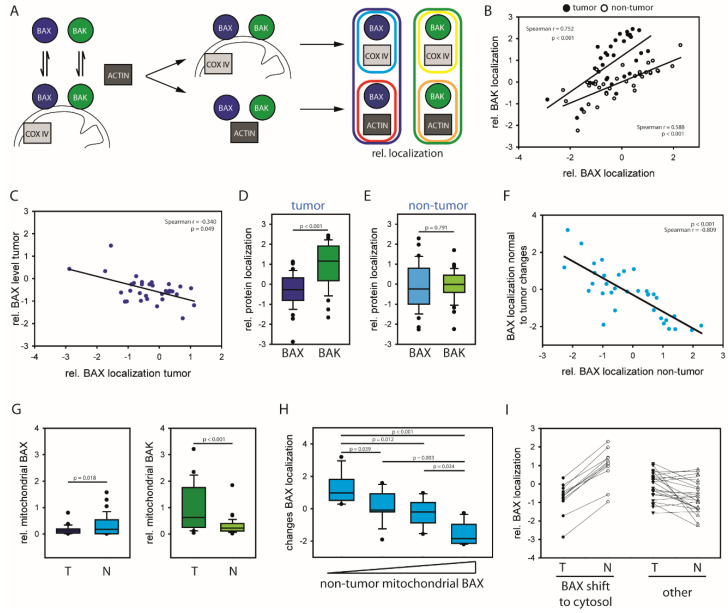
(**A**) BAX and BAK constantly translocate to the mitochondria and retrotranslocate back into the cytosol. Localization equilibriums are characterized by determining relative mitochondrial protein (rel. mitochondrial BAX (cyan), mitochondrial BAX (blue circle) per COX IV (light gray rectangle), rel. mitochondrial BAK (yellow), mitochondrial BAK (green circle) per COX IV (light gray rectangle)), and relative cytosolic protein (rel. cytosolic BAX (red), combining cytosolic BAX (blue cycle) with β-ACTIN (dark gray rectangle); rel. cytosolic BAK (orange), combining cytosolic BAK (green cycle) with β-ACTIN (dark gray rectangle)). Relative protein localization (BAX: blue; BAK: green) is the quotient of relative mitochondrial protein and relative cytosolic protein. High relative protein localization values indicate shift towards the mitochondria independent of the cellular protein concentration. (**B**) Relative BAX localization vs. relative BAK localization (double log10 scale) for 34 HCC patient samples (full circle) and corresponding non-tumor samples (open circle). R- and *p*-values according Pearson’s correlations. (**C**) Relative BAX localization in tumor tissue vs. relative BAX level in tumor samples (double log10 scale) for 34 patient samples. R- and *p*-values according Pearson’s correlations. (**D**) Relative BAX/BAK localization in 34 HCC patient tumor tissues (log10 scale). *p*-value according to *t*-test. (**E**) Relative protein localization (log10 scale) of BAX and BAK in 34 HCC patient non-tumor tissue samples. *p*-value according to *t*-test. (**F**) Relative BAX localization in non-tumor tissue vs. changes in relative BAX localization between non-tumor and tumor samples from 34 HCC patients (double log10 scale). R- and *p*-values according Pearson’s correlation. (**G**) Relative mitochondrial BAX (left) and BAK (right) levels for tumor (T) and non-tumor (N) samples from 34 HCC patients. *p*-value according to *t*-test. (**H**) Changes from non-tumor to tumor relative BAX localization of 34 HCC categorized according to the mitochondrial BAX levels in non-tumor tissue (N = 8 for extreme levels each and N = 9 for moderate levels each). *p*-value according to One Way ANOVA using Holm-Sidak method. (**I**) Relative mitochondrial BAX levels for tumor (T) and non-tumor (N) samples from 11 HCC patients with pronounced cytosolic BAX shift from non-tumor to tumor samples compared to samples without this shift (N = 23).

**Figure 2 cancers-12-01437-f002:**
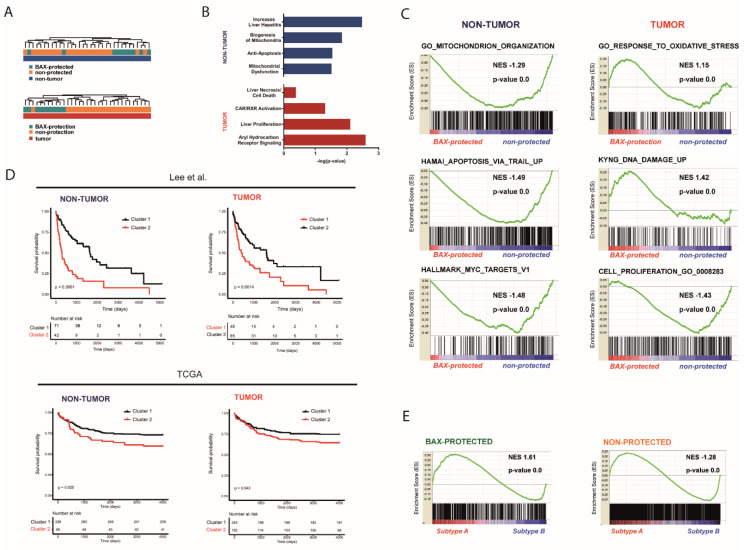
(**A**) Unsupervised clustering analyses based on genes differently regulated between BAX-protection and non-protection subgroups in non-tumorous and tumor tissue. (**B**) Activated signaling pathways between BAX-protected and non-protected subgroups in non-tumorous and tumor tissue identified by Ingenuity pathway analyses. (**C**) Gene set enrichment analysis (GSEA) for non-tumorous and tumor tissue. Normalized enrichment score (NES) reflects degree of overrepresentation for each group at the peak of the entire set. Statistical significance calculated by nominal *p*-value of the ES by an empirical phenotype-based permutation test. (**D**) Kaplan–Meyer analyses based on the specific transcriptome profiles in non-tumorous and tumor tissue using public available data from authentic human HCC of 139 patients from Lee et al. and of 395 patients from the TCGA database. (**E**) Gene set enrichment analysis (GSEA) for BAX-protected and non-protected tumor associated gene expression signatures during malignant transformation on prognostic HCC subgroups (panels A and B). Normalized enrichment score (NES) reflects degree of overrepresentation for each group at the peak of the entire set. Statistical significance calculated by nominal *p*-value of the ES by an empirical phenotype-based permutation test.

**Figure 3 cancers-12-01437-f003:**
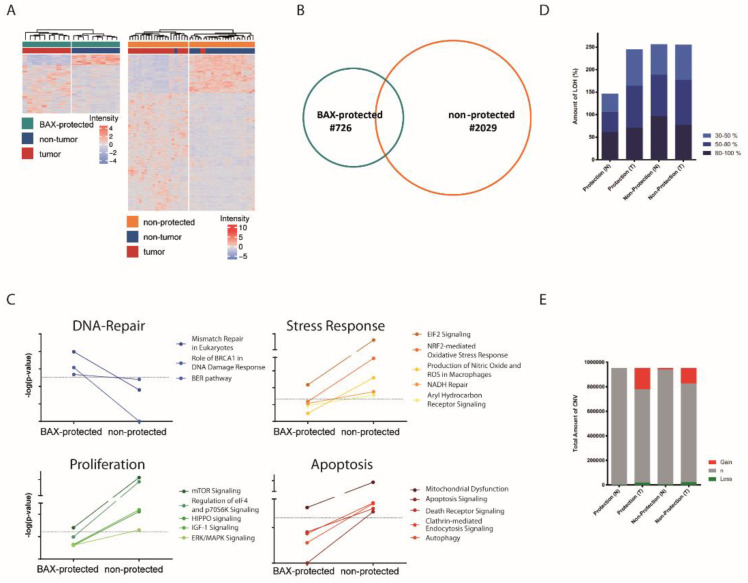
(**A**) Unsupervised clustering analyses based on corresponding genes differently regulated between non-tumorous and tumor tissue in BAX-protected and non-protected subgroups. (**B**) Venn diagram demonstrating the overlap of differentially regulated genes in BAX-protected and non-protected tumor subgroups during malignant transformation. (**C**) Activated signaling pathways during malignant transformation examined by comparative ingenuity pathway analyses based on BAX-protected and non-protected tumor subgroup transcriptome profiles. (**D**) Presence of loss of heterozygosity (LOH) in percentage based on SNP array analyses for BAX-protection and non-protection in non-tumorous and tumor tissue. (**E**) Copy number alterations based on SNP array analyses for BAX-protected and non-protected tumor subgroup in non-tumorous and tumor tissue.

**Figure 4 cancers-12-01437-f004:**
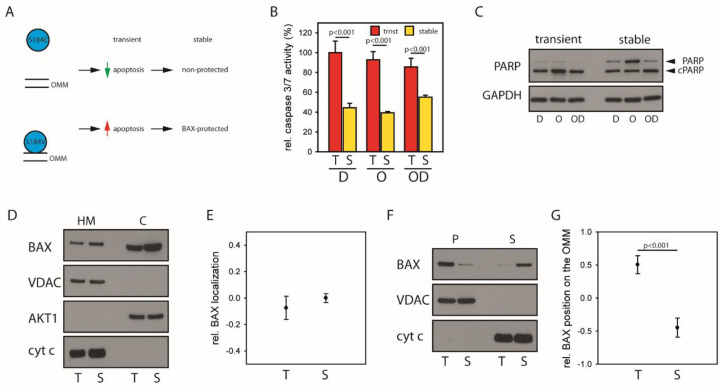
(**A**) Cells were challenged either with mainly cytosolic BAX (S184E, top) or predominantly mitochondrial BAX (S184V, bottom). The experiments were designed to increase the apoptosis predisposition through a large mitochondrial BAX pool, testing the hypothesis that cells challenged with predominantly mitochondrial BAX select for genetic alterations associated with BAX-protected tumors. (**B**) Caspase 3/7 activity induced by 1 µM Daunorubicin (D), 30 µM Olaparib (O) or the combination (OD) for 24 h in HCT116 BAX/BAK DKO cells with either transient (red, trnst, T) or stable (yellow, stable, S) expression of predominantly mitochondrial BAX S184V. Data ± SEM. N = 3 and *p*-values according to one-way ANOVA using the Holm–Sidak method. Data adjusted to BAX S184V expression. (**C**) Western blot analysis of caspase substrate PARP cleavage in HCT116 BAX/BAK DKO cells with either transient or stable expression of predominantly mitochondrial BAX S184V as in panel (**G**) following treatment with 1 µM Daunorubicin (D), 30 µM Olaparib (O), or the combination (OD). N = 3. (**D**) Subcellular localization of wild type BAX transiently (T, 4 h) or stably (S) expressed in HCT116 BAX/BAK DKO cells after fractionation analyzed by Western blot. Separation of cytosol (C) and heavy membrane fraction (HM, mitochondria) was controlled using AKT1 and VDAC or cyt c, respectively. N = 3. (**E**) Relative wild type BAX localization (log10 scale) after transient (T) and stable (S) expression in HCT116 BAX/BAK DKO cells analyzed by fractionation depicted in D. (**F**) Carbonate extraction (pH 11.5) of wild type BAX transiently (T, 4 h) or stably (S) expressed in HCT116 BAX/BAK DKO cells analyzed by Western blot. Separation of supernatant (S) containing OMM-associated proteins and pellet (P) with OMM-integral proteins was controlled using cyt c (released during the procedure) and VDAC, respectively. N = 3. *p*-value according to *t*-test. (**G**) Relative wild type BAX distribution on the OMM between OMM-integral and OMM-associated protein pool determined by carbonate extraction (log10 scale) after transient (T) and stable (S) expression in HCT116 BAX/BAK DKO shown in F.

**Figure 5 cancers-12-01437-f005:**
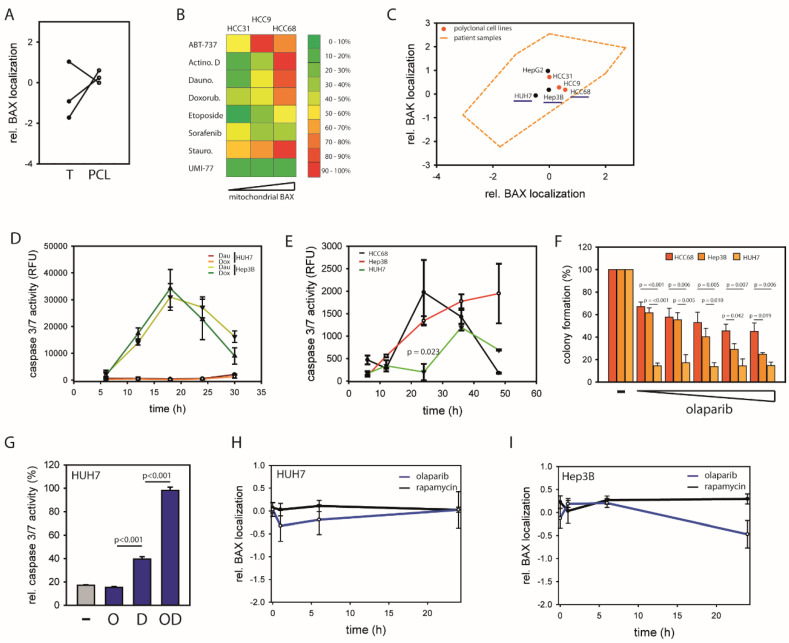
(**A**) Relative BAX localizations of polyclonal cell lines (PCL, HCC9, HCC31, and HCC68) and the corresponding tumor samples (T) are displayed in log10 scale. (**B**) Comparison of apoptotic response of PCL (HCC9, HCC31, and HCC68) after 24 h treatment with ABT-737 (1 µM), ActinomycinD (ActinoD, 1 µM), Daunorubicin (Dauno, 1 µM), Doxorubicin (Doxorub, 5 µM), Etoposide (100 µM), Sorafenib (5 µM), Staurosporine (Stauro, 1 µM), or Umi-77 (1 µM) according to caspase 3/7 activation. (**C**) Relative BAX localization vs. relative BAK localization (double log10 scale) for HepG2 cells, HUH7 cells and Hep3B cells in black. For comparison area of relative localizations in 34 HCC patient samples (orange broken line) and 3 polyclonal cell lines (red) are displayed. HUH7 cells, Hep3B cells and HCC68 cells (underlined in blue) share a similar BAK localization but show the full range of different BAX localizations in the tested cell lines. (**D**) Caspase 3/7 activity in HUH7 cells and Hep3B cells in response to 1 µM Daunorubicin or 5 µM Doxorubicin treatment for 24 h. Data ± SEM. N = 3. Apoptotic response of both cell lines after treatment with ABT-737 (1 µM), ActinomycinD (ActinoD, 1 µM), Etoposide (100 µM), Sorafenib (5 µM), Staurosporine (Stauro, 1 µM), or Umi-77 (1 µM) in Supporting Appendix A. (**E**) Caspase 3/7 activity induced in HUH7 (green) Hep3B (red) and HCC68 cells (black) in response to 5 µM Rapamycin. Data ± SEM. N = 3. (**F**) Clonogenic survival of HCC68 (red), Hep3B (orange) or HUH7 (yellow) following incubation with 0, 10, 20, 30, 40, or 50 µM Olaparib for 72 h. Data represent averages ± SEM. N ≥ 3 and *p*-values according to one-way ANOVA using the Holm–Sidak method. (**G**) Caspase 3/7 activity induced by 30 µM Olaparib (O), 1 µM Daunorubicin (D) or the combination (OD) in HUH7 cells. Data ± SEM. N = 3. (**H**) Changes of relative BAX localization of HUH7 cells with either Olaparib (black) or Rapamycin (gray) within 24 h. Data represent averages ± SEM. N ≥ 3 (**I**) Effect of Olaparib (black) or Rapamycin (gray) on relative BAX localization of Hep3B cells within 24 h. Data represent averages ± SEM. N ≥ 3.

**Table 1 cancers-12-01437-t001:** Top molecular and cellular functions identified by Ingenuity Pathway Analysis in the protection and non-protection subgroup during malignant transformation.

**Non-Protected**		
**Functions**	***p*-Value Range**	**#Molecules**
Protein Synthesis	−0.3 × 10^−3^–1.17 × 10^−13^	255
Amino Acid Metabolism	−0.57 × 10^−3^–1.77 × 10^−11^	42
Small Molecule Biochemistry	−0.57 × 10^−3^–1.77 × 10^−11^	285
Lipid Metabolism	−0.57 × 10^−3^–7.14 × 10^−8^	214
Molecular Transport	−0.57 × 10^−3^–7.14 × 10^−8^	262
**BAX-Protected**		
**Functions**	***p*-Value range**	**#Molecules**
RNA Post-Transcriptional Modification	−0.28 × 10^−2^–6.00 × 10^−5^	10
DNA Replication, Recombination and Repair	−0.17 × 10^−2^–1.46 × 10^−4^	12
Cell Morphology	−0.17 × 10^−2^–4.75 × 10^−4^	23
Cellular Function and Maintenance	−0.17 × 10^−2^–4.75 × 10^−4^	18
Cell Cycle	−0.17 × 10^−2^–4.90 × 10^−4^	2^4^

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
