# Peer review of "BAX Redistribution Induces Apoptosis Resistance and Selective Stress Sensitivity in Human HCC"

_cancers, 2020, doi:10.3390/cancers12061437_

Round 1
Reviewer 1 Report
The revised manuscript looks good to me.
Reviewer 2 Report
All comments have been addressed
This manuscript is a resubmission of an earlier submission. The following is a list of the peer review reports and author responses from that submission.
Round 1
Reviewer 1 Report
Despite good arguments, the work needs improvement about the presentation of the results and their description in order to be published. In addition, the discussion needs to be deepen to propose some perspectives
- In Fig-3H: what reason the author choose the treatment with Rapamycin, Daunorubicin, Olaparib or the combination in HCC cells
- Authors should provide quantification method of clonogenic survival assay
- The author only test the activation of caspase3/7/PARP were examine the cell death, this evidences is insufficient, it should be use the Annexin V/PI staining by flow cytometry.
- In line 310-311; the author provide the BAX-protected tumors induces chromosomal instability with subsequent oncogenic dependence on DNA repair, it’s should be provide strong evidences to examine this results.
- The image of figure is poor resolution, such as Fig-2 and Fig-3
Reviewer 2 Report
Funk et al. use tumor and non-tumor tissue samples from hepatocelluar carcinoma (HCC) patients to study HCC’s response to apoptosis-inducing drugs. This study analyzes the HCC samples by examining the subcellular localization of Bax and Bak, two functionally similar apoptosis effector proteins that are known to shuttle between the cytosol and mitochondria. They found that in tumor samples, but not the non-tumor counterparts, Bax is preferentially cytosolic. This observation led to their classification of the HCC samples into two groups, the “Bax-protected” tumors, which display predominantly cytosolic Bax localization, and the “non-protected” tumors, which show a predominantly mitochondrial presence of Bax. Based on this classification, the authors performed gene expression profiling studies in both the tumor and non-tumor samples to identify distinct gene expression patterns in Bax-protected vs non-protected tumors. They find that non-protected tumors had activation of metabolic processes and proteins synthesis involving apoptosis, proliferation and stress response pathways. In Bax-protected tumors, while apoptotic signaling was downregulated, activation of DNA replication, recombination, and repair pathways was increased. This information was applied to two independent cohorts of HCC patients and used to demonstrate that the Bax-protected phenotype is associated with poor clinical outcome. In a clever design, the authors expressed a mitochondrially targeted mutant of Bax, BaxS184V, in Bax/Bak DKO HCT116 cells, and the survivors of this apoptotic challenge confirmed the Bax-protected gene profile. Importantly, they demonstrated the sensitivity of Bax-protected tumor cell lines to DNA damaging agents, correlating with their phenotype of having increased activity of DNA repair pathways. Overall this study gives evidence for a link between Bax/Bak localization and tumorigenesis as well as sensitivity to apoprotic stimuli, and provides new thinking on the treatment of HCC patients according to the subcellular localization of Bax.
I have a few questions and suggestions:
- Most figures, including the labeling, do not have the minimal resolution, making it impossible to fully appreciate the data.
- Why does Bax seem to be bound in the cytosol in certain tumors? What is the mechanism for this? Are there mutations in Bax, changes on the mitochondria or differential regulation among the Bcl-2 family proteins? Does the DNA repair pathways have anything to do with this or are they just a downstream result of this and why? These details seem to be missing and could be included in the discussion.
- Figure 1 shows Bak localization in non-tumor samples is similar to Bax and mitochondrial Bak is low in non-tumor tissue. These seems to go against common knowledge. Is this only due to shuttling between the mitochondria and cytoplasm?
- For Fig3 G and H there is no untreated or DMSO control treated group. ABT+UMI-77 combination treatment should be used to show activity of these Bax mutants.
- For the transient expression of BaxS184V and S184E a western is shown in S10. Is this expression similar in the selected and stably expressed S184V pools? A side by side comparison would be informative.
- The legend in Figure 3 references Rapamycin treatment for H. (line 214) but this in not shown. It seems to be only included in the supplemental figures (remove this from the legend).
- What is the treatment time for 3G and 4G? This is not stated in the figure legends or the materials and methods.
- Figure 4 legend references Figs S9-11 (line 261) this should reference S16-18 instead
- The axis for caspase 3/7 activity measured by RFUs varies so much between Fig 4D, S16, S17 and 4E and S18. Fig 4E may look similar to HUH7 results in 4D if the axes were the same.
- S18 shows single treatment of either ABT-737 or UMI-77. The combination of these treatments would be a good control to show Caspase3/7 activity.
- The materials and methods list Cisplatin and Rapamycin treatment for clonogenic survival assays (line 402-403). The data for these are not shown; remove from methods. Also the Olaparib treatment for clonogenic survival is listed as 72 hrs in the materials and methods and 24 hours in Figure 4 legend. Which treatment time was used?
Reviewer 3 Report
In presented manuscript authors describe the interesting results of analysis of Bax and Bak in samples of tumor and non-tumor cell isolates from patients with hepatocellular carcinoma (HCC). They found that tumour isolates differ in Bax localisation, indicating that in some of them cell are also protected from Bax cell death-inducing activity. They also characterised differences between these isolates and found downregulation of apoptosis signalling pathways and up regulation of DNA repair pathways in isolates that are protected from Bax, as well as different sensitivity to several inhibitors (of Bax-protected vs. non protected cell). Taken together, their data indicate that Bax localization in tutors is linked with characteristics of tumors including apoptosis resistance and reflecting the differential mechanisms of malignant transformation and may be an important factor tutor treatment.
All experiments are well designed and executed. Data are well interpreted and conclusions are reasonable.
In my opinion, the weakness of this manuscript is in the language. Most of the Results chapter is described in wery complicated way and for me was very hard to follow. In description of deifferences in Bax/Bak localisation I was never sure, what is compered with what and what shifts were. I find even calling cell Bax-protected and non-protected confusing.
Although I found results very interesting and important and I reccomend to accept this manuscript for publication in Cancers, I would recommend to rewrite the results section in a way that would be easier to follow.